# Evaluating Spatial-Temporal Clogging Evolution in a Meso-Scale Lysimeter

**Jui-Hsiang Lo, Qun-Zhan Huang** , **Shao-Yiu Hsu** \*, **Yi-Zhih Tsai** **and Hong-Yen Lin**

Department of Bioenvironmental Systems Engineering, National Taiwan University, Taipei 10617, Taiwan
\* Correspondence: syhsu@ntu.edu.tw

**Abstract:** When surface water infiltrates soil, the fine soil particles carried in water gradually clog soil pores and form a low-permeability soil layer. Clogging impacts the variations in pore water pressure heads in soil and effective hydraulic conductivity. However, few studies have connected field measurements of pore water pressure heads to clogging in soil. This study proposed a diagram to demonstrate the relationship between the normalized pore water pressure head ($\lambda$) and effective hydraulic conductivity ($K_{eff}$) based on a conceptual 1-D vertical infiltration model. The coevolution of $\lambda$ and $K_{eff}$ indicated the occurrence of clogging and its location relative to the pore-pressure measurement point. We validated the $\lambda$-$K_{eff}$ diagram based on a series of numerical simulations of infiltration experiments in a lysimeter. The simulation results showed that the proposed diagram not only indicated the occurrence of clogging but also the development of the unsaturated zone beneath the upper clogging layer. Furthermore, we used a diagram to analyze the spatiotemporal changes in permeability in a lysimeter during three cycles of physical infiltration experiments. The experimental data presented with $\lambda$-$K_{eff}$ diagram indicated cracking on the soil surface, and clogging gradually developed at the bottom of the lysimeter.

**Keywords:** low-impact development; bio-retention; infiltration; clogging





## 1. Introduction

Clogging can be induced by sedimentation of fine-grained particles and biochemical processes [1]. When surface water with suspended fine particles infiltrates the soil, the particles gradually clog the soil and form a low-permeability layer, specifically known as the clogging layer [2–5]. Moreover, soil biochemical cycling significantly affects terrestrial and aquatic ecosystems and water quality in agriculture. Biochemical cycling can also influence the pollutant removal capability of soil (e.g., metals, phosphorus, and nitrogen). However, bioreaction of soil organic matter (e.g., nitrate) [1] may increase the biomass in the soil and induce bioclogging. Bioclogging can increase the heterogeneity of porous media through various biochemical processes. Newcomer et al. [6] combined the biomass growth rate and dynamic permeability models and indicated that bioclogging can significantly control the infiltration flux. Furthermore, Ulrich et al. [7] illustrated that bioclogging is a crucial determinant of hydraulic conductivity. The effective hydraulic conductivity in the clogging zone can range from $10^{-3}$ (m/s) to lower than $10^{-8}$ (m/s). Therefore, clogging can reduce the effective hydraulic conductivity of soil and affect its infiltration capability and behavior.

The infiltration, which is controlled by the change in the hydraulic gradient, impacts the development of the unsaturated zone and surface–subsurface water interactions. Ulrich et al. [7] utilized electrical resistivity tomography (ERT) to determine the development of a pumping-induced unsaturated zone below the riverbed surface. Furthermore, when clogging gradually forms on the streambed or beneath the streambed, the infiltration rate starts to decrease and induces the development of an unsaturated zone and a drop in the subsurface water table (e.g., groundwater table). When further decreases in the

groundwater table no longer significantly affect the infiltration rate, a disconnection between surface water and subsurface water occurs [8–11]. The thickness of the clogging layer and the ratio of the hydraulic conductivities of the clogging layer to those of the aquifer are related to the occurrence of disconnection [8]. Rivière et al. [12] investigated the influence of a clogging layer on infiltration through indoor experiments and numerical simulations. The authors indicated that, at the connected stage, the infiltration rate is dependent on the hydraulic gradient between the stream and aquifer, whereas, in the fully disconnected state, the infiltration rate is constant.

The clogging layer is an important factor in irrigation water management and stormwater control engineering. The loss of irrigation water is strongly related to the clogging layer, which is a low-permeability soil layer in fields. Jha et al. [13] conducted 72 double-ring infiltrometer tests and used five infiltration models to evaluate infiltration behavior in agricultural areas. They indicated that the infiltration characteristics in the fields were related to cultivation practices, macropores, and the low-permeability soil layer. In water paddy fields, a subsurface clogging layer called the plough sole layer is required to manage the irrigation water. The surface soil and plough sole layer may crack when continuously exposed to sunlight during the dry season. Liu et al. (2004) [14] indicated that infiltration may significantly increase when cracks penetrate the ploughed soil. However, when infiltration begins, the infiltrated water carries fine soil particles into the soil. Fine soil particles gradually clog the cracked soil layer and form a new clogging layer to reduce the infiltration, percolation, and seepage rates, thereby impeding the loss of irrigation water.

Bioretention cells are popular facilities for stormwater treatment. These cells can be used to regulate flood peaks and eliminate pollutants such as heavy metals using a mixed soil media filter. However, stormwater usually introduces many soil particles into the bioretention cell and induces filter clogging. Clogging reduces the permeability of the cell, impacting biochemical processes [15,16]. Clogging on the surface of the bioretention cell significantly reduces infiltration, whereas clogging at the bottom of the bioretention cell reduces drainage.

Therefore, detecting the occurrence and location of clogging is a major task in ensuring the optimal functioning of bioretention cells [17]. Estimating the spatiotemporal evolution of clogging and unsaturated zone development is also important for studies of clogging in surface–subsurface water interactions, irrigation water management, and bioretention. Nevertheless, this usually requires the integration of field measurements of the pore water pressure head, boundary conditions, and numerical simulations. Therefore, a practical tool is needed to quickly estimate and demonstrate the evolution and location of clogging in the field.

In this study, we developed a diagram of the normalized pore water pressure head ($\lambda$) and the effective hydraulic conductivity ($K_{eff}$) (the $\lambda$-$K_{eff}$ diagram) to estimate the spatiotemporal evolution of the clogging layer for vertical one-dimensional infiltration. The $\lambda$-$K_{eff}$ diagram shows the spatio-temporal evolution of various clogging scenarios, including bottom clogging, upper clogging, and mixed clogging, which result in simultaneous upper and bottom clogging. To demonstrate the implementation of the proposed diagram, we applied it to analyze a series of measured data from an outdoor mesoscale lysimeter during different infiltration experiments. In the experiments, the infiltration flux was controlled by known pressure head conditions on the surface and bottom of the lysimeter. Furthermore, we conducted numerical simulations using the COMSOL Multiphysics® to validate the diagram and simulate the temporal changes in the pore water pressure heads and infiltration flux in the abovementioned experiments. Darcy's law and Richards' porous media flow modules were used in the numerical simulations. The $\lambda$-$K_{eff}$ diagram was successfully validated by a numerical simulation. The $\lambda$-$K_{eff}$ diagram appropriately describes the occurrence of bottom clogging in a lysimeter. Moreover, simulation results indicated that the $\lambda$-$K_{eff}$ diagram indicates not only the evolution of the upper clogging, but also the growth of the unsaturated zone induced by the upper low permeable layer in the lysimeter.

## 2. Materials and Methods

### 2.1. $\lambda$-$K_{eff}$ Diagram

Based on the 1-D vertical infiltration model (Figure 1), we developed a normalized pore water pressure head ($\lambda$) via the effective hydraulic conductivity ($K_{eff}$) diagram (i.e., the $\lambda$-$K_{eff}$ diagram) to determine the spatiotemporal evolution of clogging. In the model, $h_d$ (m), $h_s$ (m), and $h_z$ (m) are the pressure heads from the constant surface water depth, pore water pressure head at the location of the tensiometer, and constant pressure head at the bottom of the model, respectively. Furthermore, the column in the model was divided into upper and lower zones based on the location of the tensiometer. The upper zone had a length $L_{up}$ (m) and an effective hydraulic conductivity $K_{eff\_up}$, whereas the bottom zone had a length $L_{bot}$ (m) and an effective hydraulic conductivity $K_{eff\_bot}$.

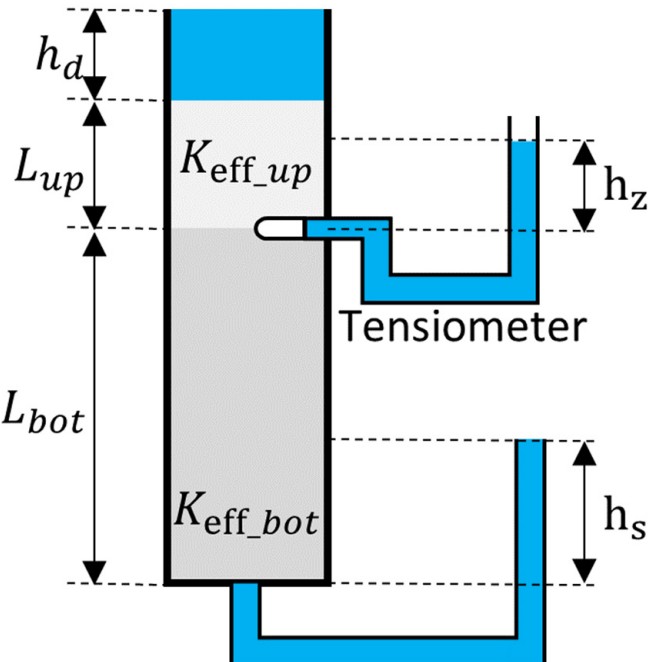

**Figure 1.** The schematic diagram of the one-dimensional vertical infiltration model. $h_d$ (m) is the constant surface water depth, $h_z$ (m) is the pore water pressure head at the location of the tensiometer, and $h_s$ (m) is the pressure head at the outlet of the model. $L_{up}$ (m) and $L_{bot}$ (m) are lengths of the upper zone and the bottom zone in the lysimeter. $K_{eff\_up}$ and $K_{eff\_bot}$ are the effective hydraulic conductivity in the upper zone and the bottom zone in the lysimeter.

Based on Darcy's law, the infiltration flux $q(t)$ (m/s) through the column can be defined as follows:

$$q(t) = -K_{eff}(t)\frac{\Delta H}{L_{up} + L_{bot}} \tag{1}$$

where $t$ (s) is time and $\Delta H = h_d + L_{up} + L_{bot} - h_s$ is the total head difference from the surface to the bottom of the column. If $q$ is negative, water infiltrates downward. The effective hydraulic conductivity of the entire column ($K_{eff}$, m/s) is:

$$K_{eff}(t) = \frac{L_{up} + L_{bot}}{\frac{L_{up}}{K_{eff\_up}(t)} + \frac{L_{bot}}{K_{eff\_bot}(t)}} \tag{2}$$

where $K_{eff\_up}$ (m/s) denotes the effective hydraulic conductivity in the upper zone of the column. $K_{eff\_bot}$ (m/s) is the effective hydraulic conductivity at the bottom zone of the column. The $K_{eff}$, based on the measured data, is calculated by

$$K_{eff}(t) = -q(t)\frac{L_{up} + L_{bot}}{\Delta H} \tag{3}$$

The temporal change of $h_z(t)$ can be expressed in the form of

$$h_z(t) = h_d + L_{up} - \frac{\Delta H(t)}{1 + \frac{K_{eff\_up}(t)}{K_{eff\_bot}(t)} \frac{L_{bot}}{L_{up}}} \tag{4}$$

According to Equation (4), when upper clogging occurs and the upper effective hydraulic conductivity ($K_{eff\_up}$) is reduced, the pressure head from the tensiometer ($h_z$) decreases. However, when bottom clogging occurs and decreases the bottom's effective hydraulic conductivity ($K_{eff\_bot}$), the pressure head from the tensiometer ($h_z$) increases. Here, we propose a normalized soil water pressure head $\lambda$ (Equation (5)) to express the change in pressure head ($h_z$) when clogging occurs. By substituting Equation (4) into Equation (5), the $L_{up}$, $L_{bot}$, $K_{eff\_up}$, and $K_{eff\_bot}$ are related to the variation in $\lambda$ before and after clogging in the column.

$$\lambda(t) = \frac{h_z(t) - h_z^*}{\Delta H(t)} \tag{5}$$

where $h_z^*$ (m) is the initial pore water pressure head, which was measured before any clogging occurred or at the beginning ($t = 0$) of infiltration.

The $\lambda$-$K_{eff}$ diagram (Figure 2) is proposed to describe the relationship between the normalized pore water pressure head, overall effective hydraulic conductivity, bottom effective hydraulic conductivity, and upper effective hydraulic conductivity. In Figure 2, the intersection of the red and blue solid lines indicates the initial effective hydraulic conductivity and pore water pressure head in the column, respectively. When bottom clogging occurs, the evolution of the clogging occurs in the upper area of the $\lambda$-$K_{eff}$ diagram ($\lambda > 0$). If pure bottom clogging occurs, which means that the upper effective hydraulic conductivity does not change, the evolution of the clogging follows the red solid line. In contrast, when upper clogging occurs, the evolution of the clogging is shown in the lower area of the $\lambda$-$K_{eff}$ diagram ($\lambda < 0$). If pure upper clogging occurs, which means that the bottom effective hydraulic conductivity does not change, the evolution of the clogging follows the blue solid line. When both $K_{eff\_up}$ and $K_{eff\_bot}$ decrease simultaneously, the $\lambda$-$K_{eff}$ point is located within the mixed clogging zone (grey region) between the red and blue lines. When the results are out of the grey region, it means that the $K_{eff\_up}$ and $K_{eff\_bot}$ are larger than their initial values.

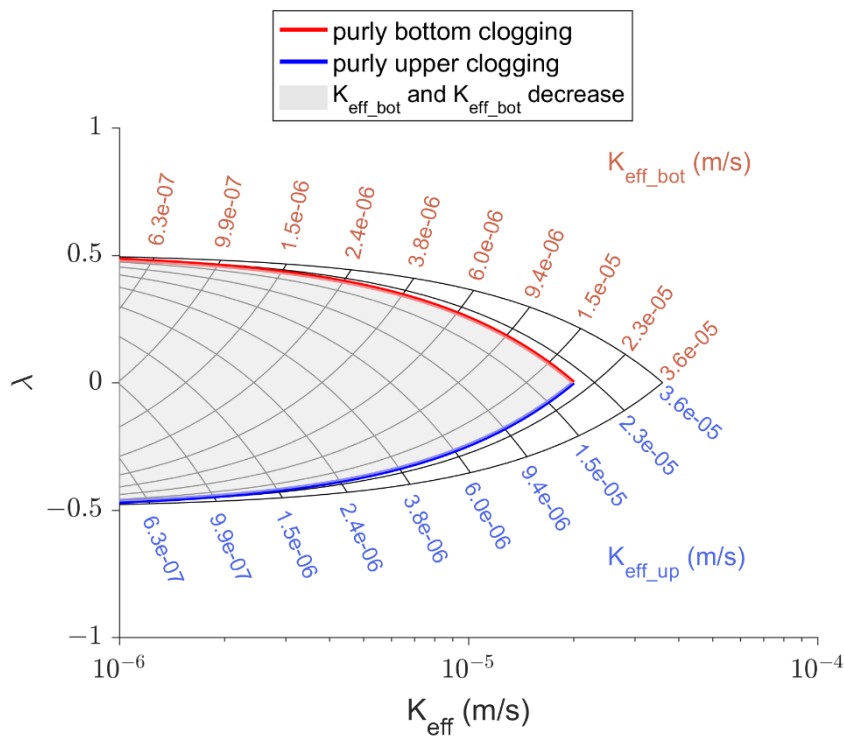

**Figure 2.** An example of the $\lambda$-$K_{eff}$ diagram. The red solid line indicates the development of pure bottom clogging. The blue solid line indicates the evolution of pure upper clogging. The region that is encircled by red and blue solid lines denotes mixed clogging (i.e., upper clogging and bottom clogging occurring simultaneously). $K_{eff\_up}(t = 0) = 2.00 \times 10^{-5}$ (m/s). $K_{eff\_bot}(t = 0) = 5.49 \times 10^{-5}$ (m/s). $L_{up} = 0.93$ (m). $L_{bot} = 0.9025$ (m).

## 2.2. Case Study with Lysimeter Experiments

### 2.2.1. Setup of Lysimeter Experiments

We used previous chronological infiltration experiments in three cycles with different stainless-steel pipe heights (Table 1). The open-air lysimeter (Figure 3) in this study was built using impermeable concretes that were 4 m long, 1.5 m wide, and 2.3 m deep (overflow notch side). A permeable pipe with a length of 4 m and a 15.24 cm radius was buried at the bottom of the lysimeter for drainage. The slope of the permeable pipe at the bottom of the lysimeter was 0.1 m/m. The permeable pipe was then wrapped with a non-woven fabric of 0.01 m thickness to prevent the loss of sand particles (Figure 3a). The hydraulic conductivity of the nonwoven fabric sheet estimated by constant-pressure head experiments ranged from $1 \times 10^{-4}$ (m/s) to $1 \times 10^{-5}$ (m/s). According to the range of the estimated hydraulic conductivity as estimated using a model [18], the pore size of the non-woven fabric sheet was 5–50 μm.

River sand and mud were used to fill the lysimeter. As shown in Figure 3b, the lysimeter was filled with river sand and a 0.1 m-thick mud layer. The river sand had an effective grain size ($D_{10} = 0.138$ mm) which was coarser than that of mud ($D_{10} = 0.017$ mm) (Figure 4). The saturated hydraulic conductivities of the river sand ($K_{rs}$) and mud ($K_m$) were $1.48 \times 10^{-4}$ m/s and $2.45 \times 10^{-6}$ m/s, respectively.

An overflow notch can maintain a constant water depth of 0.22 m on the surface ponding region during the infiltration experiments. Overflow water was collected by the overflow well. To calculate the overflow rate, we placed a water level recorder (Micro-Driver DI602, Van Essen Instruments, Delft, The Netherlands) at the bottom of the well and mounted a flow meter (ND10-TATAAA flow sensor, Aichi Tokei Denki Co., Ltd., Japan) at the outlet of the well. A flow meter with the same product model was mounted on the inlet valve to record the inflow rate. In addition, a flow meter (Karman Vortex System Flow

Sensor KSL-10L, REGAL JOINT Co., Ltd., Kanagawa, Japan) was installed at the outlet of the drainage pipe.

To control the boundary condition, we mounted and connected stainless-steel pipes at the outlet of the drainage pipe (Figure 3b). The height of the stainless-steel pipe was used to represent the pressure head at the bottom of the column ($h_s$, m) in the 1D vertical infiltration model (Figure 1). The height of the pipe could be adjusted by adjusting the number of connecting pipe sections. The usable heights of the stainless-steel pipe in the experiment were 1.5 m, 1.1 m, 0.7 m, 0 m, $-0.5$ m, and $-0.7$ m. The negative heights of the stainless-steel pipe were calculated based on the height of the outlet at a height of 0 m. The pore water pressure heads ($h_z$) in the sand in the lysimeter was measured by a tensiometer (T4e-2, UMS) which was buried in the middle of the lysimeter at a depth of 0.93 m from the surface of the mud layer. All the measurement instruments were calibrated prior to conducting the infiltration experiments.

**Table 1.** The bottom boundary conditions and experimental results in each infiltration experiment. The sequence number implied the sequence of the use of the pipe height in experimental cycles.

| Experimental Cycle | Sequence No. | Heights of the Stainless-Steel Pipe at the Outlet $h_s$, (m) | The Pore Water Pressure Head from the Tensiometer $h_z$ (m) | Infiltration Flux $q$ (m/s) |
|---|---|---|---|---|
| E1 | 1 | 1.5 | 0.850 | $-6.67 \times 10^{-6}$ |
| | 2 | 1.1 | 0.740 | $-7.95 \times 10^{-6}$ |
| | 3 | 0.7 | 0.610 | $-9.84 \times 10^{-6}$ |
| | 4 | 0.0 | 0.470 | $-1.21 \times 10^{-5}$ |
| E2 | 5 | 1.5 | 1.096 | $-5.73 \times 10^{-6}$ |
| | 6 | 1.1 | 1.001 | $-7.46 \times 10^{-6}$ |
| | 7 | 0.7 | 1.023 | $-9.01 \times 10^{-6}$ |
| | 8 | 0.0 | 1.002 | $-1.10 \times 10^{-5}$ |
| E3 | 9 | 1.5 | 0.970 | $-4.10 \times 10^{-6}$ |
| | 10 | 0.7 | 0.870 | $-7.13 \times 10^{-6}$ |
| | 11 | 0.0 | 0.730 | $-7.70 \times 10^{-6}$ |
| | 12 | $-0.5$ | 0.537 | $-7.90 \times 10^{-6}$ |
| | 13 | $-0.7$ | 0.452 | $-8.25 \times 10^{-6}$ |

2.2.2. Process of Infiltration Experiments with the Lysimeter

We conducted chronological infiltration experiments in three cycles with different heights of the stainless-steel pipe (Table 1). Each experiment started with a 1.5 m height stainless-steel pipe and then stepwise reduced to the next specified height, as listed in Table 1. The difference between the overflow and inflow rates was used to estimate the infiltration flux ($q$, m/s). The drainage rate was determined by the measured flow rates at the outlet valve. If the lysimeter reaches a steady-state, the infiltration flux ($q$) is equal to the drainage rate.

Before conducting infiltration experiments, the lysimeter was saturated by turning on the inlet and simultaneously turning off the outlet valve. The lysimeter reached saturation when the measured pore water pressure from the tensiometer was equal to 1.15 m, which is the maximum depth of the tensiometer below the water surface. After the lysimeter reached the steady-state, we mounted a 1.5 m-height stainless-steel pipe at the outlet valve and turned on the outlet valve to start the infiltration experiment. The height of the stainless-steel pipe was adjusted when the infiltration rate ($q$) and measured pore water pressure head ($h_z$) reached a stable condition. The measured infiltration flux ($q$, m/s) and pore water pressure head ($h_z$, m) from the tensiometer are listed in Table 1.

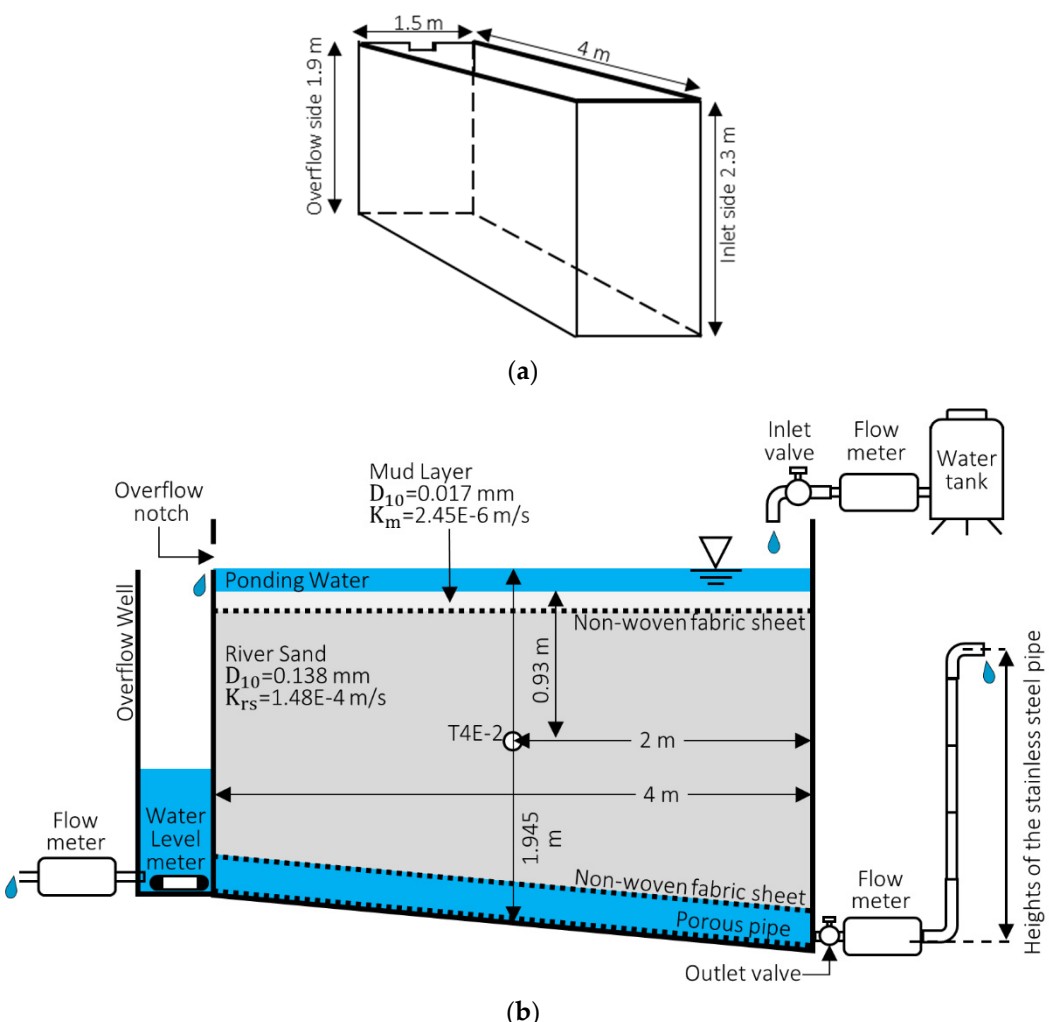

**Figure 3.** A schematic diagram of the lysimeter. (**a**) Perspective view. (**b**) Section view and the equipment set.

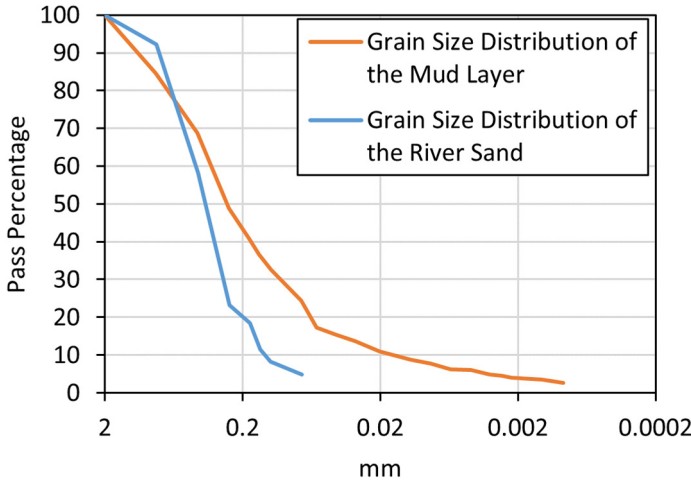

**Figure 4.** The particle size distributions of the river sand and the mud layer.

### 2.3. The Lysimeter Simulation

COMSOL Multiphysics® 5.5, which is an advanced finite-element numerical simulation software, was used to simulate the pore water pressure head in each experiment. The

shape and size of the lysimeter and the soil properties were determined based on the experimental design (Figure 3). The constant pressure head on the mud layer surface ($h_d$, m) was given to 0.22 m. The constant pressure heads ($h_s$, m) at the outlet valve for each simulation scenario are listed in Table 2. The rest of the boundaries of the lysimeter were defined as no-flow boundaries. Darcy's law and Richards' interfaces for the porous-media flow module were used in the numerical simulations. In Darcy's law interface, the governing equation is Darcy's law combined with the continuity equation and expressed as:

$$\frac{\partial(\rho\varepsilon)}{\partial t} + \nabla \cdot (\rho u) = Q_m \tag{6}$$

where $\mu$ is the dynamic viscosity of the fluid (Pa·s). The water content $\varepsilon_p$ is equal to the porosity because of the fully saturated porous medium. $p$ is the pore pressure (Pa), $\rho$ is the density of the fluid (kg/m$^3$), and $Q_m$ is the source term (kg/(m$^3$s)). The Darcy velocity $u$ (m/s) is:

$$u = -\frac{\kappa_s}{\mu}(\nabla p + \rho g \nabla z) \tag{7}$$

where $\kappa_s$ denotes the permeability of the saturated porous medium (m$^2$). $\kappa = K\mu/\rho g$ in where $K$ is the saturated hydraulic conductivity at the interface of Darcy's Law. For Richards' interfaces, $\kappa_s$ is replaced by the relative permeability ($\kappa_r$, m$^2$), which is a function of the effective saturation ($S_e$) and can be calculated from the soil water characteristics curve (SWCC). The $\varepsilon_p$ could be less than that of the porosity. The governing equation is as follows:

$$\rho\left(\frac{C_m}{\rho g} + S_e S\right)\frac{\partial p}{\partial t} + \nabla \cdot \rho\left(-\frac{\kappa_s}{\mu}\kappa_r(\nabla p + \rho g \nabla z)\right) = Q_m \tag{8}$$

where $C_m$ is the specific moisture capacity (m$^{-1}$), $S$ is the storage coefficient (Pa$^{-1}$), and $z$ is the vertical elevation (m).

Table 2 presents the simulation results for the pore water pressure head and the infiltration flux for each scenario. The simulation scenarios listed in Table 2 included pure bottom clogging (S1, S2, and S3) and upper unsaturated zone scenarios (S4). The Darcy's Law porous media flow module was applied in simulation scenarios S1, S2, and S3. The pressure heads at the bottom of the lysimeter ($h_s$, m) in S1, S2, and S3 were the same as those in experimental cycles E1, E2, and E3. The saturated hydraulic conductivity ($K_{nw}$) of the non-woven material in the drainage pipe was manually reduced from $6.4 \times 10^{-7}$ to $8.37 \times 10^{-8}$ (m/s) in the numerical models. The reduction in the saturated hydraulic conductivity resulted in bottom clogging. The saturated hydraulic conductivities of the river sand ($K_{rs}$) and mud layer ($K_m$) were given as $1.48 \times 10^{-4}$ m/s and $2.45 \times 10^{-6}$ m/s.

Once an upper low-permeability layer, that is, the mud layer, occurs owing to clogging, an unsaturated zone can develop just beneath the mud layer when the groundwater level is lowered, and air is allowed to move into the upper soil. Therefore, in the upper unsaturated zone scenario (S4), we showed the development of an unsaturated zone just beneath the mud layer owing to upper clogging. $K_m$, $K_{rs}$, and $K_{nw}$ were $2.45 \times 10^{-6}$ m/s, $1.48 \times 10^{-4}$ m/s and $6.4 \times 10^{-7}$ m/s, respectively. Thus, we used Richards' porous media flow module to simulate the development of an unsaturated zone. To build the numerical model based on Richards' equation, the soil water characteristics curves (SWCC) are required. Nevertheless, only the SWCC of the river sand was necessary because of the full saturation of the upper clogging layer during infiltration. Therefore, we conducted pressure plate tests to parameterize the SWCC of river sand in the van Genuchten model (van Genuchten, 1980), which can be expressed in the form of

$$\theta(h) = \theta_r + \frac{\theta_s - \theta_r}{\left[1 + (\alpha h)^n\right]^{1-1/n}} \tag{9}$$

where $h$ (m) is the soil pore water pressure head and $\theta$ (m$^3$/m$^3$) is the estimated water content. The optimized saturated water content $\theta_s$ (m$^3$/m$^3$) was 0.38. The optimized residual water content $\theta_r$ (m$^3$/m$^3$) was 0.048. The optimized parameters $\alpha$ (1/m) and $n$ were 3.55 (1/m) and 3.45, respectively.

**Table 2.** Simulation results of the pore water pressure head at the location of the tensiometer ($h_z$) and the infiltration flux ($q$) in each simulation scenario.

| Scenarios | COMSOL Module | $h_s$ (m) | $K_m$ (m/s) | $K_{rs}$ (m/s) | $K_{nw}$ (m/s) | $h_z$ (m) | $q$ (m/s) |
|---|---|---|---|---|---|---|---|
| S1 | Darcy | 1.5 | $2.45 \times 10^{-6}$ | $1.48 \times 10^{-4}$ | $6.40 \times 10^{-7}$ | 0.74 | $-8.79 \times 10^{-6}$ |
| | | 1.1 | $2.45 \times 10^{-6}$ | $1.48 \times 10^{-4}$ | $3.27 \times 10^{-7}$ | 0.63 | $-1.12 \times 10^{-5}$ |
| | | 0.7 | $2.45 \times 10^{-6}$ | $1.48 \times 10^{-4}$ | $2.66 \times 10^{-7}$ | 0.50 | $-1.40 \times 10^{-5}$ |
| | | 0.0 | $2.45 \times 10^{-6}$ | $1.48 \times 10^{-4}$ | $1.89 \times 10^{-7}$ | 0.35 | $-1.73 \times 10^{-5}$ |
| S2 | Darcy | 1.5 | $2.45 \times 10^{-6}$ | $1.48 \times 10^{-4}$ | $2.51 \times 10^{-7}$ | 0.87 | $-6.05 \times 10^{-6}$ |
| | | 1.1 | $2.45 \times 10^{-6}$ | $1.48 \times 10^{-4}$ | $2.00 \times 10^{-7}$ | 0.75 | $-8.69 \times 10^{-6}$ |
| | | 0.7 | $2.45 \times 10^{-6}$ | $1.48 \times 10^{-4}$ | $1.57 \times 10^{-7}$ | 0.67 | $-1.04 \times 10^{-5}$ |
| | | 0.0 | $2.45 \times 10^{-6}$ | $1.48 \times 10^{-4}$ | $1.22 \times 10^{-7}$ | 0.54 | $-1.31 \times 10^{-5}$ |
| S3 | Darcy | 1.5 | $2.45 \times 10^{-6}$ | $1.48 \times 10^{-4}$ | $2.40 \times 10^{-7}$ | 0.88 | $-5.90 \times 10^{-6}$ |
| | | 0.7 | $2.45 \times 10^{-6}$ | $1.48 \times 10^{-4}$ | $1.41 \times 10^{-7}$ | 0.70 | $-9.71 \times 10^{-6}$ |
| | | 0.0 | $2.45 \times 10^{-6}$ | $1.48 \times 10^{-4}$ | $9.89 \times 10^{-8}$ | 0.63 | $-1.13 \times 10^{-5}$ |
| | | −0.5 | $2.45 \times 10^{-6}$ | $1.48 \times 10^{-4}$ | $8.49 \times 10^{-8}$ | 0.57 | $-1.24 \times 10^{-5}$ |
| | | −0.7 | $2.45 \times 10^{-6}$ | $1.48 \times 10^{-4}$ | $8.37 \times 10^{-8}$ | 0.54 | $-1.32 \times 10^{-5}$ |
| S4 | Richards | 1.5 | $2.45 \times 10^{-6}$ | $1.48 \times 10^{-4}$ | $6.40 \times 10^{-7}$ | 0.74 | $-8.79 \times 10^{-6}$ |
| | | 1.1 | $2.45 \times 10^{-6}$ | $1.48 \times 10^{-4}$ | $6.40 \times 10^{-7}$ | 0.42 | $-1.17 \times 10^{-5}$ |
| | | 0.7 | $2.45 \times 10^{-6}$ | $1.48 \times 10^{-4}$ | $6.40 \times 10^{-7}$ | 0.02 | $-1.18 \times 10^{-5}$ |
| | | 0.0 | $2.45 \times 10^{-6}$ | $1.48 \times 10^{-4}$ | $6.40 \times 10^{-7}$ | −0.31 | $-1.17 \times 10^{-5}$ |

## 3. Results

### 3.1. Measured Data in the Lysimeter $q$ and $h_z$

Figure 5 shows the experimental infiltration flux ($q$) and pressure head from the tensiometer ($h_z$) in Table 1. In all the experimental cycles, the drop in the pressure head at the bottom boundary ($h_s$) resulted in a decrease in $h_z$ and an increase in the infiltration flux ($q$). Nevertheless, the slope of $q(h_z)$ in E2 was steeper than that of other experimental cycles. The steeper slope indicates that the pressure head loss in E2 was less than that in E1 and E3, which caused the same infiltration rate. The results for E2 imply that the level of clogging, especially in the upper region, was weaker than that in the other experiments. In E3, the slope of $q(h_z)$ becomes gentle after $h_s < 0.7$ m. The gentle slope of $q(h_z)$ indicates that a serious loss of head was needed to result in the same infiltration flux ($q$), and the level of clogging was more serious than that in other experimental cycles.

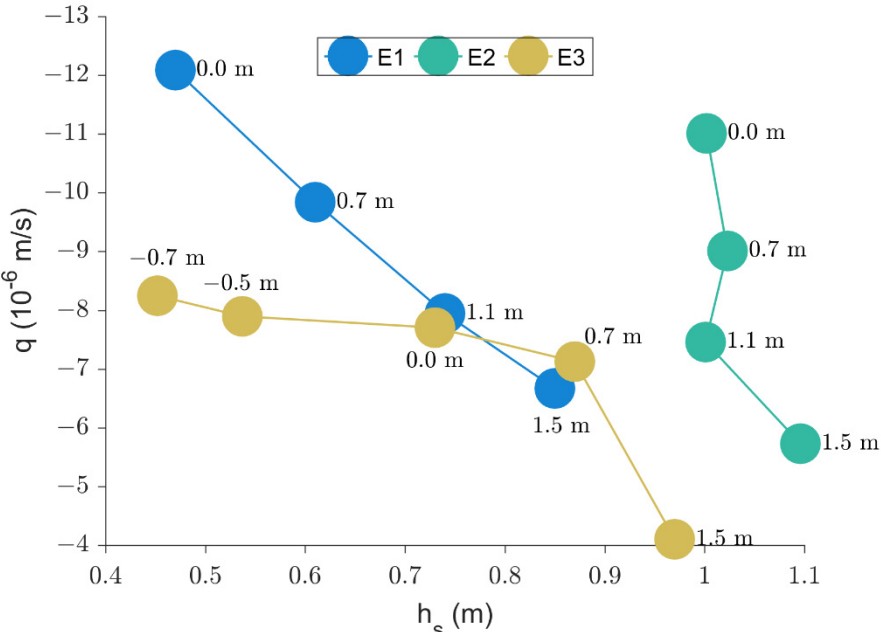

**Figure 5.** Experimental infiltration flux (q) and pore water pressure head from the tensiometer ($h_z$). The negative infiltration flux indicates the downward infiltration direction. The numbers near each dot present the pressure head at the bottom boundary ($h_s$).

### 3.2. Experimental Data in the $\lambda$-$K_{eff}$ Diagram and Diagram Validation

Figure 6 shows the normalized pressure head and the overall, bottom, and upper effective hydraulic conductivities, which were calculated using Equations (3) and (5) in the experiments (E1 to E3) and simulations (S1 to S4). The original $K_{eff\_up}$ that was used is $2 \times 10^{-5}$ (m/s), which was calculated based on the thickness of the mud layer and the depth of the tensiometer, which were $K_m = 2.45 \times 10^{-6}$ (m/s), $K_{rs} = 1.48 \times 10^{-4}$ (m/s), respectively. The applied $K_{eff\_up}$ was $5.49 \times 10^{-5}$, and was calculated using $L_{bot} = 0.9025$ (m), the assumed $K_{nw} = 1 \times 10^{-7}$ (m/s), and a 1-cm-thick $K_{nw}$ layer.

In Figure 6a–c, the simulation results of S1, S2, and S3 (green circles) follow the solid red curve and show that the development of pure bottom clogging along the nonwoven material on the drainage pipe was detected by the diagram. Figure 6d shows the simulation results for S4. The variation from point 1 to point 3 closely followed the pure upper clogging line, which is denoted by the blue solid line in the $\lambda$-$K_{eff}$ diagram. However, point 4 exhibited a reduction in effective bottom hydraulic conductivity ($K_{eff\_bot}$). This may be due to the mixing of the upper and bottom clogging or an unsaturated zone beneath the mud layer induced by an upper low-permeability layer. When the unsaturated zone reached the bottom zone, the effective hydraulic conductivity of the bottom started to decrease. Therefore, the proposed $\lambda$-$K_{eff}$ diagram can use the variations of $\lambda$ and $K_{eff}$ to determine the temporal change of $K_{eff\_bot}$ and $K_{eff\_up}$. Thus, the proposed $\lambda$-$K_{eff}$ diagram can describe the $\lambda$ and $K_{eff}$ from the lysimeter experiment and indicate the change in $K_{eff\_bot}$ and $K_{eff\_up}$.

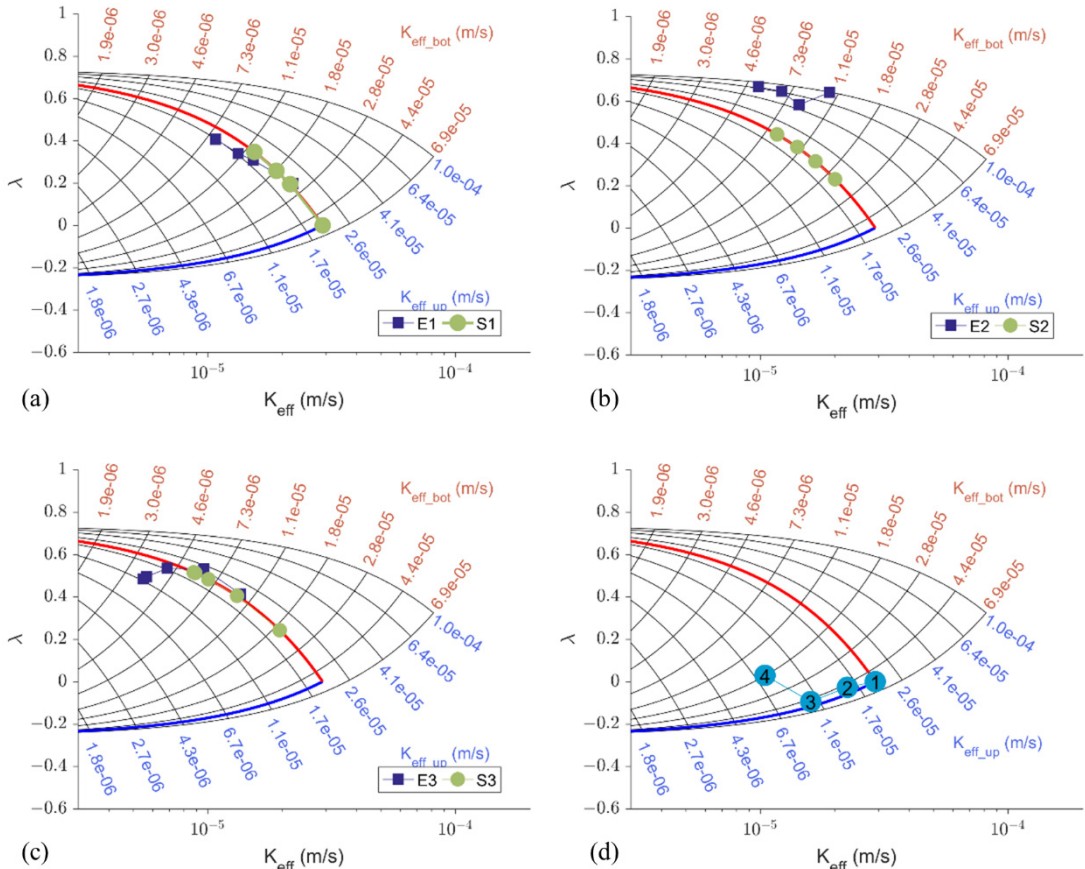

**Figure 6.** The results of (**a**) S1, (**b**) S2, (**c**) S3, and (**d**) S4. In (**d**), the numbers the circles 1, 2, 3, and 4 are the result of the simulation at $h_z$ = 1.5, 1.1, 0.7, and 0.0 m, respectively.

## 4. Discussions

### 4.1. $\lambda$ and $K_{eff\_bot}$ from Experiments and Simulation

As shown in Figure 6a–c, comparing the experimental measurements (blue squares) with the simulation results, the experimental results in E1 and E3 were highly similar to those in S1 and S3, which presented the evolution of the bottom clogging. However, the experimental results for E2 showed a much higher upper effective hydraulic conductivity than the simulation results for S2 and the red solid line. This may be because the surface mud layer cracked and shrank during the long dry period between experimental cycles E1 and E2. Surface cracks increased the permeability of the mud layer and provided preferential flow paths for infiltration. Nevertheless, although the experimental results of E2 showed higher upper effective hydraulic conductivities than the simulation results of S2, the overall variation of the normalized pore water pressure heads and effective hydraulic conductivities in E2 were still in the bottom clogging zone of the $\lambda$-$K_{eff}$ diagram. Therefore, we speculate that the primary evolution of clogging in the lysimeter was bottom clogging, and the change in the upper effective hydraulic conductivity was limited.

### 4.2. Increasing $\lambda$ and Reducing $K_{eff\_bot}$ in Each Experimental Cycle

The $\lambda$-$K_{eff}$ behavior in each cycle is opposite to the possible $\lambda$-$K_{eff}$ behavior when the upper layer is less permeable than the lower layer. In a more permeable subsoil, the saturated hydraulic conductivity is affected by the upper low-permeability layer [19]. For instance, the unsaturated zone occurred and increased with a reduction in $h_s$ in the S4 simulation. When the soil is unsaturated, the saturated hydraulic conductivity is lower than that under saturated conditions. It is expected that as the $\Delta H$ increases, the $K_{eff}$ increases and $\lambda$ decreases. However, $\lambda$ increased for each experimental cycle in this study.

The reversal behavior was caused by the covering water layer and impermeable concrete, which limited air intrusion into the soil. As there was no air intrusion, the soil maintained its hydraulic conductivity under saturated conditions. Moreover, comparing the experimental results from E1, E2, and E3 to the simulation results in S1, S2, and S3 (Figure 6a–c), bottom clogging should be significant in the experiments. Small particles may have been deposited at the bottom of the lysimeter and on the drainage pipe, and even on the broken porous pipe.

Although the soil at the bottom of the lysimeter was not sampled after the experiments, our speculation about the breakage of the porous pipe was supported by experimental findings when we dug all soil out of the lysimeter after the experiments (Figure 7). The non-woven material on the drainage pipe was covered and clogged with fine river particles. In addition, the pipe was crushed, but the crushing did not result in a gradual $K_{eff\_bot}$ decrease. The crushed pipe and sealed nonwoven material may be the two main reasons for the reduction in $K_{eff\_bot}$ in the experiments.

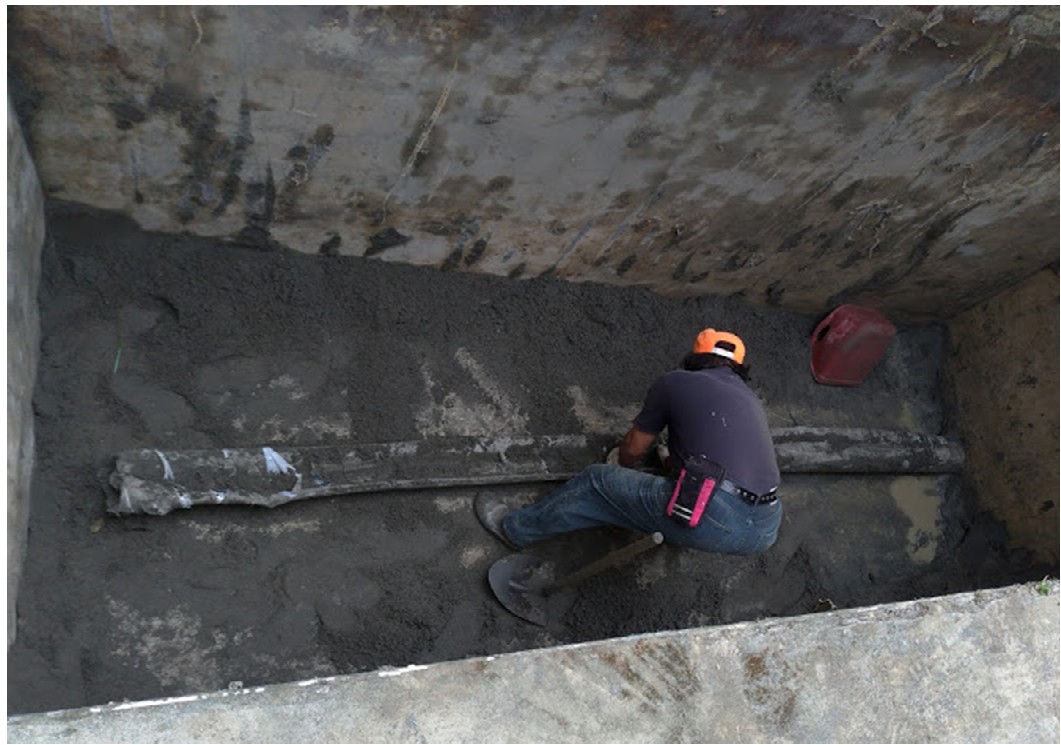

**Figure 7.** The broken drainage pipe after experiments.

### 4.3. Recovery of $K_{eff\_bot}$ When Performing the Next Cycle

Figure 8 presents all the experimental λ and $K_{eff}$ values with the λ-$K_{eff}$ diagram. The changes in λ and $K_{eff}$ from E1 to E3 show that the $K_{eff\_bot}$ did not continuously decrease. The level of clogging recovered at the beginning of E2 and E3. In other words, the initial $K_{eff}$ and $K_{eff\_bot}$ in E2 (sequence No. 5) were larger than those of sequence No. 4 in E1. The initial $K_{eff}$ and $K_{eff\_bot}$ values in E3 (sequence No. 9) were higher than those of sequence No. 8 in E2. The recovery of clogging might occur due to evaporation or changes in the pore water pressure. Regarding the former, evaporation in the soil can induce upward water flow. Ranjbaran and Datta [20] showed that the internal flow induced by evaporation of a droplet on a leaf transports bacteria to the air–water–solid contact line. In this study, the evaporation-induced flow in the break between the experimental cycles might have been able to upwards transport the tiny soil particles, and therefore, recover $K_{eff\_bot}$.

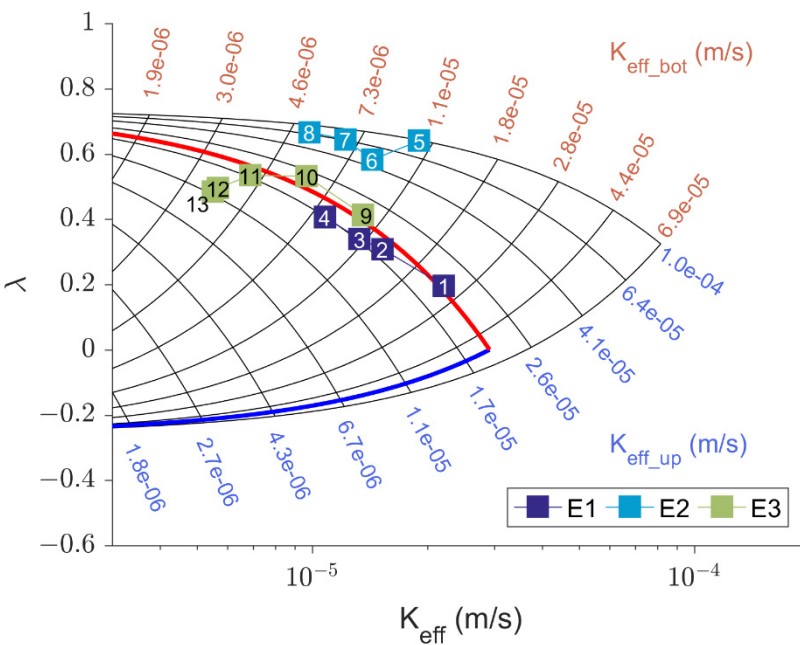

**Figure 8.** Experimental results in the $\lambda$-$K_{eff}$ diagram. The number in each square presents the experiment sequence number in Table 1.

A decrease in the pore water pressure increases the effective stress and, therefore, reduces the void ratio [21–23], which means that the pore size shrinks. Small pores usually have low permeability [18]. The $h_s$ cycled from E1 to E3 may therefore induce a repeated decrease and recovery of permeability. In our experiments, however, the beginning $K_{eff\_bot}$ in E2 or E3 is less than the beginning $K_{eff\_bot}$ in its previous cycle. The permeability of the bottom zone gradually decreased during the experimental cycle. However, owing to the heavier weight loading in a deep soil position, the reduction in the void ratio could be larger than that at a shallow position. This is another possible reason for the $K_{eff}$ reduction, mainly owing to the $K_{eff\_bot}$ reduction (see Figure 8).

*4.4. Unsaturated Zone Development*

The simulation results of S4 show the development of an unsaturated zone owing to upper clogging. Figure 9a shows the S4 simulation results using the $\lambda$-$K_{eff}$ diagram. Figure 9b–e present the variation of water content in 1.5 m, 1.1 m, 0.7 m, and 0.0 m pressure heads ($h_s$). The pressure heads ($h_s$) were controlled using the corresponding height of the stainless-steel pipe at the outlet of the drainage pipe in the lysimeter. The simulation results in the lysimeter plotted in the $\lambda$-$K_{eff}$ diagram followed a pure upper clogging line (blue line in Figure 9a). Moreover, when the height of the pipe was decreased from 0.7 m to 0 m, the simulation result moved towards the bottom clogging area. Based on the change in soil moisture in the lysimeter (Figure 9b–e), the change in direction was caused by the development of an unsaturated zone across the pressure measurement point. The unsaturated zone gradually expanded with the decrease in the height of the stainless-steel pipe. This development is similar to the findings in [7], which showed that the development of unsaturated zone in field resulted from the increase in total head difference. The increment in the total head difference was due to groundwater pumping or a drop in the groundwater table. In our simulation (i.e., S4), when the height of the stainless-steel pipe decreased from 1.5 m to 0.7 m, the unsaturated zone was still above the location of the tensiometer, which was the upper zone in the lysimeter. Nevertheless, when the height of the stainless-steel pipe decreased to 0 m, the unsaturated zone extended over the location of the tensiometer and entered the bottom zone in the lysimeter. Thus, the unsaturated zone below the tensiometer caused a reduction in $K_{eff\_bot}$ and induced

an increase from node 3 (Figure 9d) to node 4 (Figure 9e) in the mixed clogging zone in Figure 9.

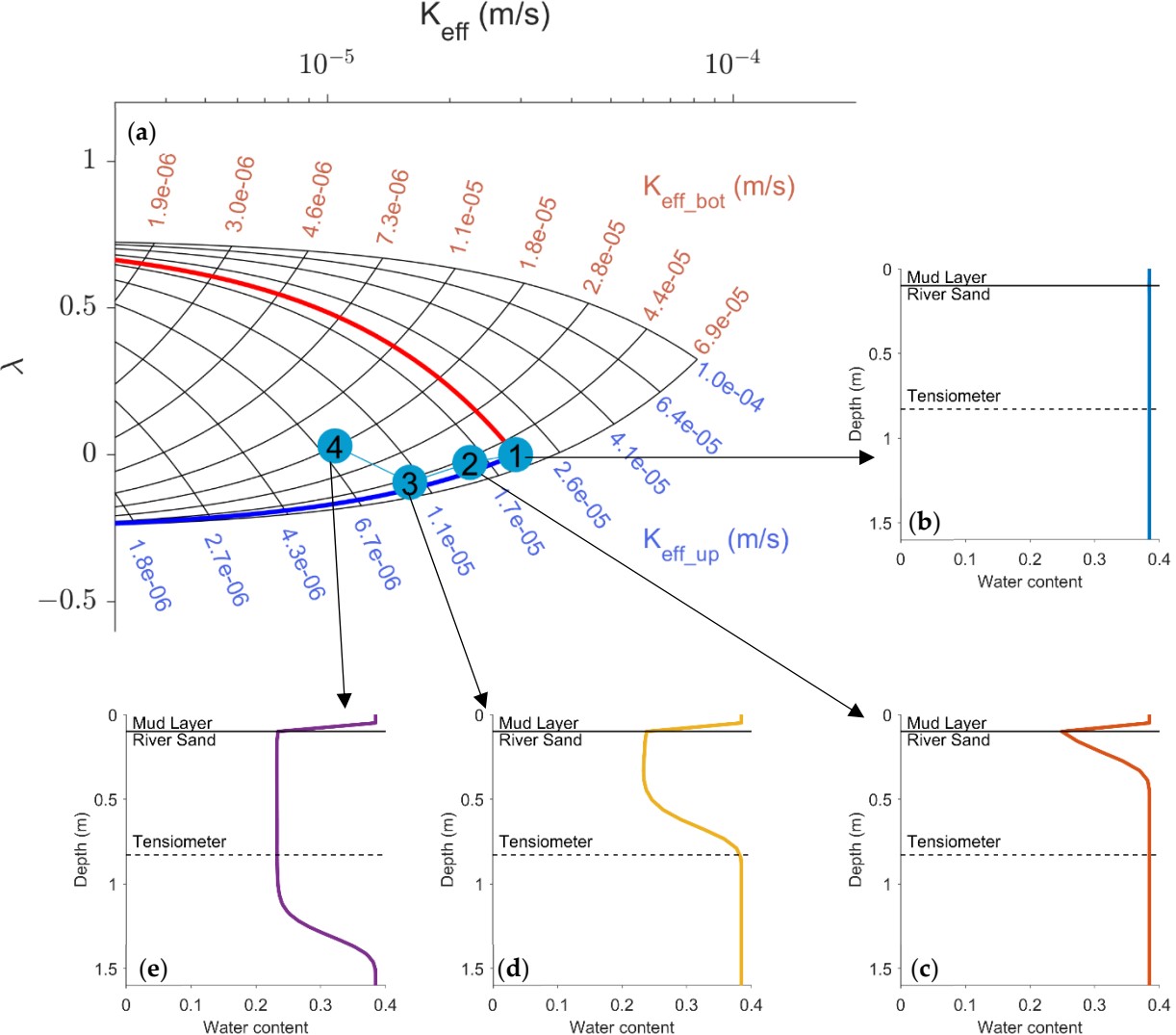

**Figure 9.** The simulation results in the upper clogging scenario. (**a**) The overall evolution of clogging in the height of stainless-steel pipe ($h_s$) with (**b**) 1.5 m, (**c**) 1.1 m, (**d**) 0.7 m, and (**e**) 0.0 m. The upper black solid lines in (**b**) to (**e**) present the height of the bottom of the mud layer, while the black dash lines show the height of the tensiometer in the lysimeter.

Without introducing fine particles into the lysimeter, clogging at upper soil layer can occur due to the formation of bioclogging, which is induced by biochemical processes. It is worth investigating the mechanism and implications of bioclogging on the changes in effective hydraulic conductivity [1,6,7]. Future studies and models could combine the biomass growth rate [6] and the change in hydraulic gradient [7], which control the infiltration flux, to elucidate the process of bioclogging at a finer scale. Such a study could also be used to demonstrate the capability of the $\lambda$-$K_{eff}$ diagram to describe the evolution and location of the clogging and unsaturated zone development induced by bioclogging.

## 5. Conclusions

The proposed $\lambda$ the $\lambda$-$K_{eff}$ diagram indicated the reduction and increase in permeability above and below the location of the tensiometer (i.e., the upper and bottom zones). Clogging occurred in the upper zone when the $\lambda$ and $K_{eff}$ simultaneously decreased. On the contrary, when $K_{eff}$ decreased and $\lambda$ increased simultaneously, bottom clogging oc-

curred. By utilizing the $\lambda$-$K_{eff}$ diagram with numerical simulations, we showed that the $\lambda$-$K_{eff}$ diagram can be appropriately used to determine the evolution of the pure bottom and upper clogging in the lysimeter by measuring infiltration flux and pore water pressure. Moreover, the simulation results with the upper clogging on the $\lambda$-$K_{eff}$ diagram presented the influence of upper clogging on the development of the unsaturated zone.

In our case study, the results from each experimental cycle indicated bottom clogging in the lysimeter. Bottom clogging may have been caused by the clogging of the non-woven material on the porous drainage pipe. The crushed drainage pipe could have also contributed towards the $K_{eff\_bot}$ reduction. The recovery of $K_{eff\_bot}$ was observed at the beginning of the experimental cycles E2 and E3. This recovery may be related to the evaporation-induced movement of fine particles. The high-pore water pressure head at the beginning of the experiment may have enlarged the pore size and thus increased $K_{eff\_bot}$. However, further studies are required to investigate the influence of particle movement and pore water pressure on permeability. In the experimental cycle E2, the $\lambda$-$K_{eff}$ diagram showed that $K_{eff\_up}$ was larger than that in E1 and E3. The high $K_{eff\_up}$ in E2 was due to the occurrence of cracks on the surface of the mud layer.

The unsaturated zone scenario in the lysimeter was investigated using the S4 simulation. In S4, an increase in the total head difference enlarged the unsaturated zone beneath the mud layer, congruent with findings of a previous study [7]. The expansion of the unsaturated zone reduced the upper effective hydraulic conductivity ($K_{eff\_up}$) and normalised the pore water pressure head ($\lambda$). When the unsaturated zone crossed the location of the tensiometer, $K_{eff\_up}$ decreased and $\lambda$ increased.

Overall, this study links the pore water pressure head and infiltration flux measurements to the detection of spatial-temporal clogging evolution in the lysimeter. Furthermore, we proposed a novel model and the corresponding $\lambda$-$K_{eff}$ diagram to provide a simple and effective tool for rapid and efficient estimation of the spatial-temporal evolution of clogging in bioretention cells.

**Author Contributions:** Conceptualization, S.-Y.H.; Formal analysis, J.-H.L. and Q.-Z.H.; Investigation, J.-H.L., Q.-Z.H., Y.-Z.T. and H.-Y.L.; Methodology, J.-H.L., Q.-Z.H., Y.-Z.T. and H.-Y.L.; Project administration, S.-Y.H.; Resources, S.-Y.H.; Software, J.-H.L. and Q.-Z.H.; Validation, J.-H.L. and Q.-Z.H.; Visualization, J.-H.L. and Q.-Z.H.; Writing—original draft, J.-H.L. and Q.-Z.H.; Writing—review & editing, Q.-Z.H. and S.-Y.H. All authors have read and agreed to the published version of the manuscript.

**Funding:** This research was funded by the National Science and Technology Council (NSTC), Taiwan, under grant MOST 111-2116-M002-025 and National Taiwan University (NTU) under grant NTU-CC-111L894701.

**Institutional Review Board Statement:** Not applicable.

**Informed Consent Statement:** Not applicable.

**Data Availability Statement:** The data are contained within this article.

**Acknowledgments:** We thank anonymous reviewers for their comments and suggestions.

**Conflicts of Interest:** The authors declare no conflict of interest.

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
