# Peer review of "Evaluating Spatial-Temporal Clogging Evolution in a Meso-Scale Lysimeter"

_land, doi:10.3390/land11091518_

Round 1

Reviewer 1 Report

Review of Evaluating spatial-temporal clogging evolution in a meso-scale lysimeter by Lo et al.

In this research, the authors developed a conceptual ????? diagram to estimate the spatio-temporal evolution of clogging based on field and numerical experiments of clogging in response to infiltration. In the field, they used a 3D lysimeter to perform experiments, and then used a 1D numerical model to perform the simulations. The set of lysimeter experiments and numerical models is appropriate to address the research topic. The novelty of this work is using the experiments and models to estimate the location of the clogging layer, please emphasize this point in the introduction and conclusion.

Introduction: In the introduction, the authors focus the problem on two types of cases where clogging matters: agriculture and bioretention. While I understand that clogging matters for these cases, it actually matters across all types of systems both natural and managed.

Instead of focusing the introduction on situations where clogging matters, I recommend the authors structure the introduction to focus on the state of the science on clogging and impacts of clogging across natural and managed systems, and to focus on WHY clogging matters.  The literature is incredibly rich and diverse on this topic and the authors devote very little space or time to the rich literature. Clogging occurs from two simultaneous and dynamic processes: 1) filling of pore-spaces by fine sediment particles, and 2) by microbial biomass and extracellular polymeric substances that develop in response to flow conditions (the role of bacteria in clogging is present in ALL systems). Please use the literature provided below to include new paragraphs on the wealth of information, theory, experiments, and models already available about clogging processes and WHY they matter:

11)      (Ulrich et al., 2015) showed the developed of an unsaturated zone in response to clogging for the first time using ERT.

22)      (Newcomer et al., 2016) provided a similar conceptual model to your ????? diagram, and similar equations that you have derived.

33)      Clogging process will have implications in agriculture, bioretention cells, and in natural systems because of the importance of sediments for biogeochemical cycling. Biogeochemical cycling is why clogging occurs, so this represents an important role of environments to remove nitrate for example. (Rogers et al., 2021)

Discussion/Conclusion: Despite the interesting results, the authors do little to put the novel results in the appropriate literature context for discussion of clogging processes. Please expand on the discussion of your results, and place your results in the context of the literature. For example, you could expand on the discussion of updated conceptual models for clogging, and you could expand on the implications for unsaturated zone development.

References

Rogers, D. B., et al. (2021). Modeling the Impact of Riparian Hollows on River Corridor Nitrogen Exports. Frontiers in Water3, 590314. https://doi.org/10.3389/frwa.2021.590314

Newcomer, M. E., et al. (2016). Simulating bioclogging effects on dynamic riverbed permeability and infiltration. Water Resources Research52(4), 2883–2900. https://doi.org/10.1002/2015WR018351

Ulrich, C., et al. (2015). Riverbed Clogging associated with a California Riverbank Filtration System: An assessment of mechanisms and monitoring approaches. Journal of Hydrology529(3), 1740–1753. https://doi.org/10.1016/j.jhydrol.2015.08.012

Reviewer 2 Report

The authors propose a “diagram to demonstrate the relationship between the normalized pore water pressure head and effective hydraulic conductivity based on a conceptual 1-D vertical infiltration model.” However, in the description and presentation in Section 2.2, it is unclear how this diagram advances significantly beyond application of 1-D flow analysis for layered soil systems that is covered in most soil mechanics textbooks. The authors need to make a stronger case for the novelty of the proposed diagram.

Important details were missing regarding the materials used in the lysimeter experiments. For the river sand and “mud”, only the D10 values were reported. However, other characteristics of the particle-size-distributions (PSDs) beyond D10 can be important indicators of clogging risk. The authors should provide the PSDs of both materials to allow for replication of the experiments. In addition, the dry (bulk) densities of the sand and mud were not reported, even though density significantly impacts hydraulic conductivity.

Figure 3 shows the presence of a nonwoven fabric sheet (geotextile) at the interface of the mud and sand layers. Why was this upper geotextile included if the intent was to allow clogging of the lower layer by the fine particles? Most bioretention systems do not have a geotextile at the surface, so this also does not seem representative of the described applications.

The text in section 4.2 suggests the clogging of the geotextile at the bottom at the drainage pipe was neither intentional or controlled. What was the Apparent Opening Size (AOS) of the geotextile? Was it the appropriate AOS to be expected to be compatible with the sand? If the authors did not want the fabric to clog with “fine river sandy particles” than why not just select the correct geotextile to avoid the issue?

In section 4.1 the authors “speculate” about the evolution and location of clogging with depth in the lysimeter. However, collection of soils samples with depth after the experiment and subsequent particle size distribution testing could have easily verified or refuted the speculated clogging evolution. It is unclear why the authors did not take this final step to verify the results and strengthen the impact of their study.

Author Response

We appreciate the comments from the reviewer and have revised our manuscript accordingly. All revisions to the manuscript are highlighted in yellow. Please check the attached doc file for our point-by-point responses to each comment. 

Reviewer 3 Report

Manuscript number: land-1819004

Article Type: Article

Evaluating spatial-temporal clogging evolution in a meso-scale lysimeter

Jui-Hsiang Lo, Qun-Zhan Huang, Shao-Yiu Hsu, Yi-Zhih Tsai, Hong-Yen Lin

Recommendation: Accept in present form

Dear Authors,

Thank you for the opportunity to review the manuscript, " Evaluating spatial-temporal clogging evolution in a meso-scale lysimeter" for the Land journal. In this study, Authors developed a normalised pore water pressure head via the effective hydraulic conductivity diagram to estimate the spatiotemporal evolution of clogging based on the vertical one-dimensional infiltration model. Given that the topic of your manuscript centers on the concepts of water infiltration through the soil, I believe that the subject matter is in line with the scope of Land journal.

The work deals with an interesting topic and seems to be very current. Research on rainwater management, especially in urbanized areas, is very important. For this purpose, devices for water retention and infiltration into the ground, for example bioretention cells, are used. This facility can be used to regulate the flood peak and eliminate pollutants such as heavy metals using a mixed soil media filter. In the case of these devices, the phenomenon of clogging is important. Clogging on the surface of the bioretention cell significantly reduces infiltration, whereas clogging at the bottom of the bioretention cell reduces drainage. Therefore, the prevention of clogging is necessary to maintain bioretention cells. Perhaps this topic is not interesting from a scientific point of view, but the results of these studies have practical significance.

In the Introduction, the authors provided a brief research background discussing the clogging process, which occurs during the operation of groundwater infiltration equipment.

The applied methods were professional and effective in attaining the object of this work. The Discussion and Conclusions sections are supported by the information, discussed inside the manuscript. The strong point of the article is the graphic part, especially interesting charts.

The paper is well-organized, containing all of the expected components.

Author Response

Thank you for your valuable comments

Reviewer 4 Report

The text contains a number of mistakes and missing blank spaces (e. g. between value and unit). There are also some sentences and phrases that are not comprehensible. In some cases the use of English language could be improved through restructuring of sentences by a native English speaker.

The following issues should be discussed and added:

·      Page 2, Introduction, line 58: “…affects the biochemical process...” The authors should explain in which way the biochemical process is affected by clogging.

·      Page 3: The term “model” is inappropriate and should be replaced by “experimental setup” or “experimental plant”.

·      Page 6, line 172: The negative heights “…-0.5 m, and -0.7 m.” have to be explained.

·      Page 7: The theoretical background of the used simulation model(s) should be explained in more detail (e.g. Darcy’s Law, Richards’ porous media flow modules,…).

·      Page 7, lines 229 – 231: The authors should describe how the values of the parameters were determined.

·      Chapter 3 (Results and discussion) and chapter 4 (Discussion) should be brought together in one chapter and the numbering should be adjusted.

The specific comments are summarized in the attached pdf file “Specific comments_land-1819004”.

Author Response

We appreciate the comments from the reviewer and have revised our manuscript accordingly. All revisions to the manuscript are highlighted in yellow. Please check the attached file for our point-by-point responses to each comment.

Round 2

Reviewer 1 Report

The authors have made substantial edits to improve the quality of the manuscript. Thank you for completing the edits and changes I requested in my review. 

Reviewer 2 Report

The authors made a conscientious effort to address the original comments.